# An analysis method of local entropy changes from atomic fluctuations

**Takafumi Ishii[1], Takashi Kojima[2]\*, Yusuke Yasuda[3], Kazushi Fujimoto[4]**

**1** Tech Solution Department, Matlantis Corporation, Chiyoda-ku, Tokyo, Japan, **2** MI technology Groupe, AI Innovation Department, ENEOS Holdings Inc., Yokohama, Kanagawa, Japan, **3** Department of Advanced Materials Science, Graduate School of Frontier Sciences, The University of Tokyo, Kashiwa-shi, Chiba, Japan, **4** Department of Chemistry and Materials Engineering, Faculty of Chemistry, Materials and Bioengineering, Kansai University, Suita-shi, Osaka, Japan

\* kojima.takashi.246@eneos.com

## Abstract

A macroscopic variable, entropy, stands as a key representation that reveals the relationship between internal structure and material properties of elastomer equipped with entropic elasticity. The entropy of network strand in cross-linked polymer structure is a critical factor for elucidating rubber elasticity. However, calculation of local entropy variation that characterize macroscopic properties has been challenging to date. In this study, we propose a method, based on molecular dynamics, that estimates the entropy of local polymer-network structure by calculating the number of states from atomic fluctuations in phase space. Our analysis of coarse-grained molecular dynamics simulations showed that the calculated entropy agrees qualitatively not only with thermodynamic principles but also with values obtained from thermodynamic integration and related approaches. Furthermore, we calculated the entropy of cross-linked polymeric structures, which extends its applicability to measurement of both the entropy of entire system and entropy change in the local structures.

## Introduction

Rubbers and elastomers are widely used in various industrial products, including automobile tires, electrical insulation, and soft contact lenses. Their properties arise from distinct polymer network structures formed by cross-linking. It is generally recognized that the cross-linking reaction randomly occurs and subsequently generates a complex and inhomogeneous network structures. Significant efforts to enhance their material properties have been conducted through development of innovative cross-linking and blending techniques to control the structures. Furthermore, many numerical and theoretical studies have been conducted with the aim of understanding the relationship between them [1–23]. However, it remains challenging to develop fundamental improvement strategies that could help resolve the trade-off between material

**Data availability statement:** Code associated with this work, including representative examples of the input data for the proposed method, is publicly available on GitHub at https://github.com/Takashi-Kojima-S/entropy-change-from-atomic-fluctuations. Due to their large size, the complete set of molecular dynamics simulation output files cannot be deposited; only representative result files have been uploaded.

**Funding:** The author(s) received no specific funding for this work.

**Competing interests:** The authors have declared that no competing interests exist.

properties, such as elastic modulus and fracture characteristics, because a quantitatively precise theory of rubber elasticity has not been established. To comprehensively understand the relationship between the polymer network and the macroscopic properties, numerous mechanistic models have been proposed [6,24–45]. For instance, the affine network models [1–4] and phantom network models [3–8] are commonly used to estimate the polymer network structure representing cross-linking density. In these models, the cross-linking points are displaced in a manner similar to that of the overall deformation. The segments of the polymer chains, that is network strands [9,10,12,13], are uniformly deformed. Furthermore, sophisticated methods not only for measuring stress originating from the cross-linking structure during deformation but also methods for quantifying polymer network structures, such as the Scanlan-Case criterion, have been proposed [12,13]. Recently, molecular dynamics (MD) simulations have been performed to thoroughly understand these relationships [46–50]. However, no matter which method is used, directly estimating entropy, a representative state variable characterizing the rubber elasticity, remains challenging, and there are still gaps between computational findings and experimental results due to the limitation in the size of the simulation model and the achievable simulation time scale.

In the polymer network system, it is well known that the dangling chains and the network strands contribute differently to the entropy change in the system [46,47,51–53]. Therefore, the decomposition of the entropy of the system into the contribution from the substructures is essential for elucidating the relationship between them, while changes in the entropy of the entire system can be calculated from variations in internal energy relative to the work done on the system, according to the first law of thermodynamics. In the field of drug discovery, calculations of molecular level entropy based on the degree of freedom have been reported, such as Schlitter's formula, and quasi-harmonic analysis [54,55]. However, determining localized entropy remains elusive, because these approaches cannot decompose the entropy of large macromolecules, such as polymers, into entropies for individual substructures. Consequently, establishing comprehensive microstructure-property linkage remains an outstanding issue, as measurable internal energy is only a portion of the source of the macroscopic properties. To unravel the mechanisms underlying the mechanical property, it is essential to develop methods for accurately valuating entropy, a vital critical characteristic of polymer materials.

To overcome this challenging problem, we develop a novel method to derive local entropy based on local atomic fluctuations. This approach ingeniously calculates the number of states (NoS) which determines entropy for each atom within its phase space consisting of the position and momentum parameters, so called μ-space, and then integrates them to derive the total NoS of the entire system possesses, and NoS of local structures possesses. In other words, our method enables us to compute each atom's contribution to macroscopic entropy because it calculates the NoS of the entire system by summing the NoS for each atom. To the best of our knowledge, methods capable of decomposing entropy down to the atomic level have not been proposed, while instances of calculating molecular level entropy from the variance of atomic fluctuations have been reported in the field of drug discovery

[54–58]. Additionally, in the field of electrochemistry, there have been reports on methods for calculating local entropy using nonequilibrium thermodynamics, but these are limited to calculating the entropy of individual molecules and have not achieved calculations for the entropy of arbitrary substructures [59,60]. In this paper, we first meticulously explain the proposed method in detail, and then demonstrate its accuracy by comparing the entropy measured using our method with the entropies calculated based on representative techniques. Moreover, we successfully applied the method to evaluating the change in entropy of the polymer network during deformation.

## Analytical method

The key point of our proposed method is integrating the NoS for each atom in the system within its phase space. This integration leads to the NoS for the system which determines the macroscopic entropy. In other words, the macroscopic entropy determined by our method can be decomposed into the atomistic-scale level entropy, because the macroscopic entropy results from the NoS for each atom. Therefore, using our method, the entropies of substructures, e.g., dangling chains and network strands, can be derived.

According to the statistical mechanics, the Boltzmann constant of the proportional factor relates the NoS of the system with the macroscopic entropy $S$ as follows:

$$S = k_B \ln W(E, \lambda, T),$$

(1)

where $k_B$ and $W$ are the Boltzmann constant and the NoS, respectively. $W$ is the function of the energy, $E$, the stretch ratio representing the deformation state, $\lambda$, and the temperature, $T$. The change in entropy $\Delta S$ occurred by a transfer from a state $W$ to another state $W'$ is measured as follows:

$$\Delta S = k_B \ln \left( \frac{W'(E', \lambda', T')}{W(E, \lambda, T)} \right).$$

(2)

Note that we assume that the NoS of each atom is defined as the volume of the phase space of them at an arbitrary time range and the range of motion of each atom in the phase space is independent of other atoms, that is, the volume remains constant during the sampling at the same state [61].

$$W(E, \lambda, T) = \prod_{a=1}^{N} V_a,$$

(3)

here $V_a$ and $N$ are the volume of the phase space regarding the atom $a$ and the number of the atoms in the system, respectively. In this study, $V_a$ is defined as follows:

$$V_a = det \left| \delta_{ij} \boldsymbol{x_i} \otimes \boldsymbol{p_j} \right|,$$

(4)

where $\boldsymbol{x}$ and $\boldsymbol{p}$ are vectors consisting of the standard deviations of the temporal changes in position and momentum of the atom, respectively. $\delta$ stands for the Kronecker-delta. In Euclidean space, they are described as follows:

$$V_a = \prod_{i=1}^{3} x_i p_i.$$

(5)

To the efficient calculation, the positions and the momentums of a few atoms consisting of the functional group, such as methyl group and methylene group, are condensed into them of a representative point based on the mass weight, $m$, of each atom as follows:

$$x_b = \frac{\sum_l m_l x}{\sum_l m_l},$$

(6)

$$p_b = \frac{\sum_l m_l p}{\sum_l m_l},$$

(7)

where $b$ and $l$ are an index of the representative point and the index of the atom mapped to the point $b$.

## Result and discussion

We assessed the computational accuracy of the developed method for two scenarios: temperature changes under isochoric conditions and deformations under isothermal conditions. For the isochoric temperature-change case, we compared the entropies measured by the developed method with those obtained from representative molecular dynamics-based approaches, namely quasi-harmonic analysis (QHA) [58] and thermodynamics integration methos (TI). For the isothermal deformation case, we compared our results with entropies derived from the first law of thermodynamics.

### Accuracy validation-1: Entropy changes under isochoric conditions

QHA assumes that fluctuations in the motions of the system can be approximated by a Gaussian probability distribution. Frequencies $\omega$ can be calculated from the determinant as follows:

$$\det\left(M^{\frac{1}{2}}\sigma M^{\frac{1}{2}} - \frac{k_B T}{\omega^2}\mathbf{1}\right) = 0,$$

(8)

here $M$ is the mass matrix with the masses of the atoms on the diagonal and all off-diagonal elements equal to zero, $\mathbf{1}$ is the unit matrix, and $\sigma$ is the covariance matrix of the 3N Cartesian coordinates where N is the number of atoms in the considered molecule or molecules. The covariance matrix has the elements:

$$\sigma_{ij} = \left\langle \left(r_i - \overline{r_i}\right)\left(r_j - \overline{r_j}\right)\right\rangle,$$

(9)

where $r$ is the coordinate of the atom $i$. The quasi-harmonic entropy at $T$, $S_{qh}(T)$, and the entropy change, $\Delta S_{qh}$, from $T = T_1$ to $T = T_2$ are calculated from the frequencies of the harmonic oscillator as follows:

$$\Delta S_{qh} = S_{qh}\left(T_2\right) - S_{qh}\left(T_1\right)$$

(10)

$$S_{qh}(T) = R\sum_i \frac{\frac{\hbar\omega_i}{k_B T}}{\exp\left(\frac{\hbar\omega_i}{k_B T}\right) - 1} - \ln\left[1 - \exp\left(\frac{-\hbar\omega_i}{k_B T}\right)\right],$$

(11)

here $R$ and $\hbar$ are the gas constat and Planck's constant divided by $2\pi$, respectively.

The entropy changes for temperature change from $T = T_1$ to $T = T_2$ obtained by TI, $\Delta S_{TI}$, is given by:

$$\Delta S_{TI} = \int_{T_1}^{T_2} \frac{C_V(T)}{T} dT,$$

(12)

where $C_V(T)$ denotes the isochoric heat capacity at temperature $T = T$ and is given by:

$$C_V(T) = \frac{\left(\langle E(T)^2\rangle - \langle E(T)\rangle^2\right)}{\left(k_B T^2\right)}.$$

(13)

 

In this study, we employed the classical Kremer-Grest model, which is a frequently employed approach in coarse-grained molecular dynamics (CGMD) simulation for polymer investigations [62,63], then applied our method to it to validate its accuracy. The system was composed of 500 polymer chains consisting of 200 beads with the length of the cubic system being approximately $49[\sigma^3]$. The density of the cell was $0.85[\frac{m}{\sigma^3}]$. $m$ and $\sigma$ are the units of mass and length, respectively. The beads were connected by the following potential:

$$U(r) = U^{FENE}(r) + U^{LJ}(r), \tag{14}$$

where $r$ is the distance between the beads. $U^{FENE}(r)$ and $U^{LJ}(r)$ are given by following:

$$U^{FENE}(r) = \begin{cases} -\frac{1}{2}kR_0 ln\left(1-\left(\frac{r}{R_0}\right)^2\right) & , \ (r \le R_0) \\ \infty, & (r > R_0) \end{cases} \tag{15}$$

$$U^{LJ}(r) = 4\epsilon\left[\left(\frac{\sigma}{r}\right)^{12} - \left(\frac{\sigma}{r}\right)^6\right], \tag{16}$$

where $k$, $R_0$, and $\epsilon$ are the spring constant, the maximum extension of the spring, and the unit of the energy, respectively. We used the following traditional parameter set: $k = 30$, $R_0 = 3.0$, $\epsilon = 1$, $\sigma = 1$. The relaxation simulations for five systems at $T = 1.0$, 1.2, 1.4, 1.6, 1.8, 2.0 were conducted. $T$ is defined as $T = \frac{k_B}{\epsilon}$ in the Kremer-Grest model. After the equilibration process, the sampling simulations were performed for $1 \times 10^8$ steps with the timestep $\Delta t = 0.006[\tau]$ under NVT ensemble. 50,000 frames in total were sampled for each temperature every 2,000 steps. Fig 1 shows the energy time series during the sampling simulation and demonstrates that all systems were fully equilibrated.

We evaluated the dependence of NoS on the number of sampled states (i.e., sampling duration). For $T = 1.0$ and $T = 2.0$, we investigated the relationship between the number of frames used to compute the NoS and the resultant NoS for randomly selected 100 chains in the system. Each measurement was repeated five times, and the mean and variance of the NoS at each sampling duration are shown in Fig 2. At both temperatures, stable statistics were obtained after 1,000 shots, corresponding to a total sampling duration of $6[\tau]$.

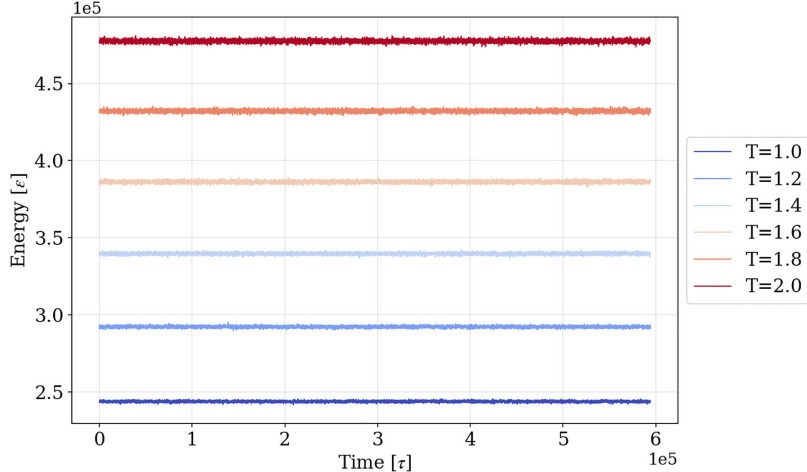

**Fig 1. The histories of energy during the sampling simulations.**

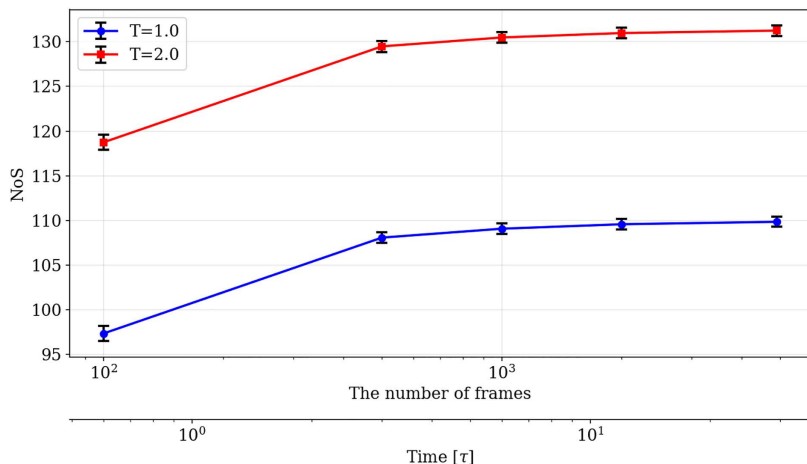

**Fig 2. Relationship between sampling duration and the NoS obtained at multiple temperatures.**

The entropy changes obtained by our method, $\Delta S_{AF}$, were compared with those measured by QHA, $\Delta S_{QH}$, and by TI, $\Delta S_{TI}$. As a reference, the entropy changes based on the $C_V(T)$ of the ideal gas, $\Delta S_{IG}$, were also calculated. Note that $C_V(T)$ of the ideal gas is $C_V(T) = \frac{3}{2}R$. The entropy changes are presented in Fig 3. Across the entire temperature range, $\Delta S_{QH}$ is the smallest. This is because $\Delta S_{QH}$ is derived solely from vibrational contributions and therefore excludes configurational and other non-vibrational components. By contrast, $\Delta S_{TI}$ is derived from the internal energy, which comprises inter-bead interaction energy and kinetic energy. Consequently, $\Delta S_{TI}$ is larger than $\Delta S_{QH}$, as it incorporates not only the vibrational contributions but also fluctuations associated with diffusive motion. Note that QHA requires significant computational cost. Therefore, to reduce computational expenses, we extracted 5 polymer chains (i.e., 1000 beads) for the calculation. The selection was repeated three times, and the mean result was used; the resulting error was less than 1%. $\Delta S_{AF}$ not only reproduces the temperature-dependent increase in entropy same as $\Delta S_{QH}$ and $\Delta S_{TI}$, but its values lie between them and are closest to the ideal-gas reference. This can be attributed to the undeformed Kremer-Grest model at a density of 0.85: the FENE-LJ bonds are unstretched and retain their equilibrium lengths, mean bond length and the maximum

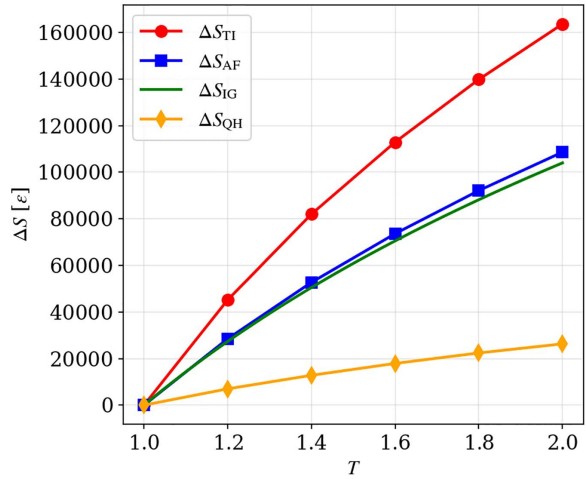

**Fig 3. The entropy changes during isochoric heating from T = 1 to T = 2.**

bond length were 0.965 and 1.088, respectively, at the end of the relaxation simulation. Thus, interparticle interactions dominate and the system behaves in a gas-like manner. These results confirm that the developed method can compute the entropy–temperature relationship with accuracy comparable to representative existing approaches.

## Accuracy validation-2: Entropy changes under isothermal conditions

As another validation, the variation in the entropy calculated using the proposed method, $\Delta S_{AF}$, was meticulously compared with the change in entropy based on thermodynamics. For this purpose, a cross-linked polymer model was constructed by performing reaction analysis on the Kremer–Grest model used in the previous validation. By conducting a reaction simulation following a relaxation simulation for $6 \times 10^7 [\tau]$, that is $1 \times 10^{10}$ steps, at T = 1.0, cross-linking bonds were created between beads within a distance threshold of 0.9, with a probability of 0.05. Two cross-linked models were created: one with 1500 cross-linking bonds and another with 2800 cross-linking bonds. The models were elongated at a speed of $4 \times 10^{-5} [\frac{\sigma}{\tau}]$ along the z-axis under NVT ensemble. Then NVT calculations were carried out at 0.25 strain increments to measure the NoS in the fixed cell length for $10,000[\tau]$. The NoS of each bead was measured using last 1,000 snapshots taken at interval of $6[\tau]$. The change in heat due to entropy variation was calculated by multiplying the obtained NoS by the Boltzmann constant and temperature.

According to the first law of thermodynamics, the change in internal energy $dU$ is expressed as the difference between the heat $dQ$ supplied to the system and the work performed by the system, as follows:

$$dU = dQ - dW. \tag{17}$$

Here,

$$dS = \frac{dQ}{T}, \tag{18}$$

$$dW = PdV. \tag{19}$$

Thus, the change in heat related to the change in entropy is described as follows:

$$TdS = dU + PdV. \tag{20}$$

Fig 4 illustrates the heat-stretch ratio relationships obtained by the proposed method and those derived from thermodynamics as mentioned above. For reference, the histories of the internal energy are also presented. The range of the stretch ratio over which the entropy obtained by the proposed method is plotted is limited to the entropic elasticity region, where the internal energy is approximately zero. The change in the heat obtained by the proposed method qualitatively matches that calculated based on thermodynamics in the entropic elasticity region. Meanwhile, both Fig 4a and b demonstrate that the heats measured by the proposed method are consistently larger than those derived from thermodynamics. These differences imply that the proposed method tends to overestimate the change in the phase space because of the assumption of the independence of the phase space for each bead, as previously described. This assumption neglects the correlation between the phase space of individual beads. In addition, the lack of equilibrium in the MD systems used to calculate the NoS could also be a contributing factor.

We confirmed that the entropy obtained by the developed method, which is based on atomic fluctuations, is comparable in accuracy to entropies computed by representative techniques; however, as noted above, the method assumes that the phase-space volume of each particle is independent of those of the other particles. To assess the validity of this assumption, we investigated the detrimental effects of correlations between neighboring particles. The 200-bead polymer

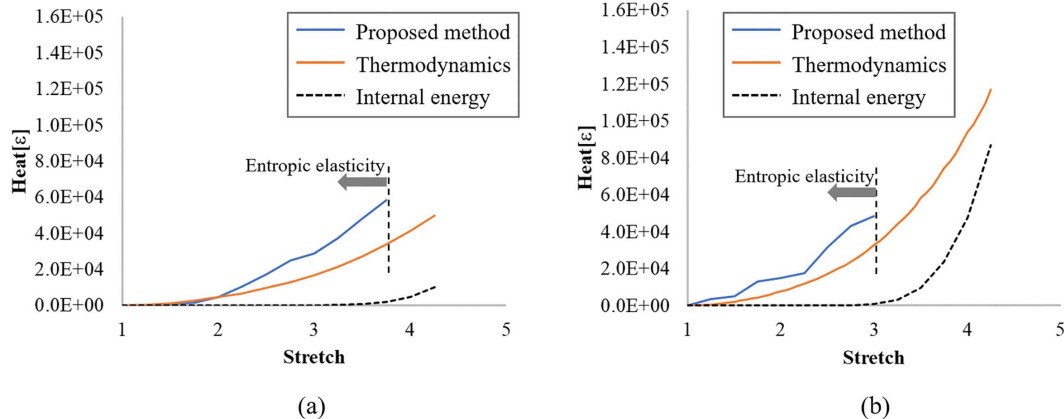

**Fig 4. Comparison between heat-stretch ratio relationships obtained by the proposed method and thermodynamics.** The history of the internal energy is also illustrated to confirm the entropic elasticity region. (a) 1500 and (b) 2700 cross-linking models.

chain used in the foregoing studies were partitioned into 2–200 segments, and we compared the segmental bonded entropy $\Delta S_{cov}$, calculated from the covariance matrix, with the multivariate total correlation, $\Delta S_{corr}$, which quantifies the entropy deficit arising from correlations. $\Delta S_{corr}$ and $\Delta S_{cov}$ are given by:

$$\Delta S_{corr} = \Delta S_{AF} - \Delta S_{cov},\tag{21}$$

$$\Delta S_{cov} = \frac{1}{2}k_B \ln\left(\prod_{i\in X}\det\left(C_i\right)\right),\tag{22}$$

where $C_i$ is the covariance matrix of segment $i$ and $X$ denotes the set of the segments. Computing the covariance matrix is computationally expensive; therefore, $\Delta S_{cov}$ was calculated using data from 100 chains same as the accuracy validation-1. Fig 5a shows that $\Delta S_{AF}$ is larger than $\Delta S_{cov}$, suggesting that $\Delta S_{AF}$ overestimates the entropy because of correlations between segments. To evaluate this hypothesis, we corrected $\Delta S_{AF}$ using the inter-segment mutual information $\Delta S_{MI}$ defined below:

$$\Delta S_{AF_{corr}} = \Delta S_{AF} - \Delta S_{MI},\tag{23}$$

$$\Delta S_{MI} = \frac{k_B}{2}\sum_{i\in X}\sum_{j\in X}\frac{\det(C_i)\det(C_j)}{\det(C_{ij})},\quad (i < j).\tag{24}$$

Fig 5a shows that the correction based on the $\Delta S_{MI}$ increases the accuracy. However, a gap remains between $\Delta S_{AF\_corr}$ and $\Delta S_{cov}$, which is likely attributable to higher-order correlations that were not fully removed by the correreation. Furthermore, the overestimation rate of the proposed method, $R_{OE}$, was calculated as follows:

$$R_{OE} = \frac{\Delta S_{AF} - \Delta S_{cov}}{\Delta S_{cov}}.\tag{25}$$

As shown in Fig 5a, for segment lengths of 10 or less, where the multivariate total correlation is small, the overestimation rate of our method was approximately twenty percents or less. This suggests that segment lengths of 10 or less exhibit characteristic dynamics distinct from both the Kuhn length and the entanglement length. At smaller length scales,

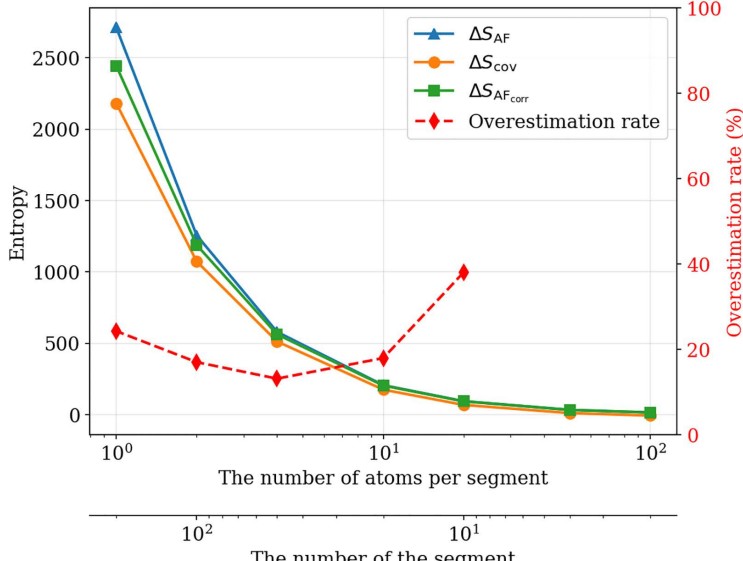

**Fig 5. Relationship between segment size and calculated entropy.** Entropy computed by the developed method, corrected entropy, and the overestimation rate of the developed method are illustrated. The segmental bonded entropy is also presented as a reference.

short-range correlations are attributable to local fluctuations arising from bond connectivity, whereas at larger length scales the correlations reflect motions of the entire chain.

## Application example: Prediction of fracture sites in cross-linked polymers

In the previous paragraph, we demonstrated that the proposed method enables to measure the macroscopic entropy by integrating the NoS of each atom. Here, we present the result of the internal structures analysis of Kremer-Grest model introduced in the previous paragraph as a case study of our method. The systems were deformed along the Z direction, and the NoS at stretch ratios of 1 and 3 were measured. As shown in Fig 4, the entropic elasticity region extends up to a stretch ratio of 3; therefore, we compared the results at a stretch ratio of 3 with the undeformed state at a stretch ratio of 1. Fig 6 shows that the comparison between the distributions of the NoS for the dangling chain and the network strand at a stretch ratio of 1 and at a stretch ratio of 3. The difference between the dangling chain and the network strand at a stretch ratio of 3 shown in Fig 6b is larger than that at a stretch ratio of 1 shown in Fig 6a, because the NoS for the network strand decreases during the deformation. This reduction is induced by the preferential extension of the network strands that follow the deformation of the system. In contrast, the NoS for the dangling chain does not significantly decrease because their flexibility prevents them from following the cell deformation. Additionally, the bimodal distribution of the network strand at a stretch ratio of 3 is induced by two types of motion: extension along the elongation direction and the compression parallel to the elongation direction.

As a next step, the NoS of specific network strands in the 1500 cross-linking model were analyzed to investigate the relationship between the change in the network strands and the failure characteristics. Atoms consisting of extended polymer chains at a stretch ratio of 5 in the energetic elasticity, where internal failure was likely to occur, were extracted based on a length threshold of 1.13. The extracted atoms shown in Fig 7a are confirmed to constitute extended polymer chains. The distributions of the NoS from a stretch ratio of 1–4 are presented in Fig 7b. It is particularly noteworthy that even though in the entropic elasticity region, where the stretch ratio is less than 2 as shown in Fig 4a, specific chains which will become extended at high stretch ratio indicating the energetic elasticity are preferentially extended and

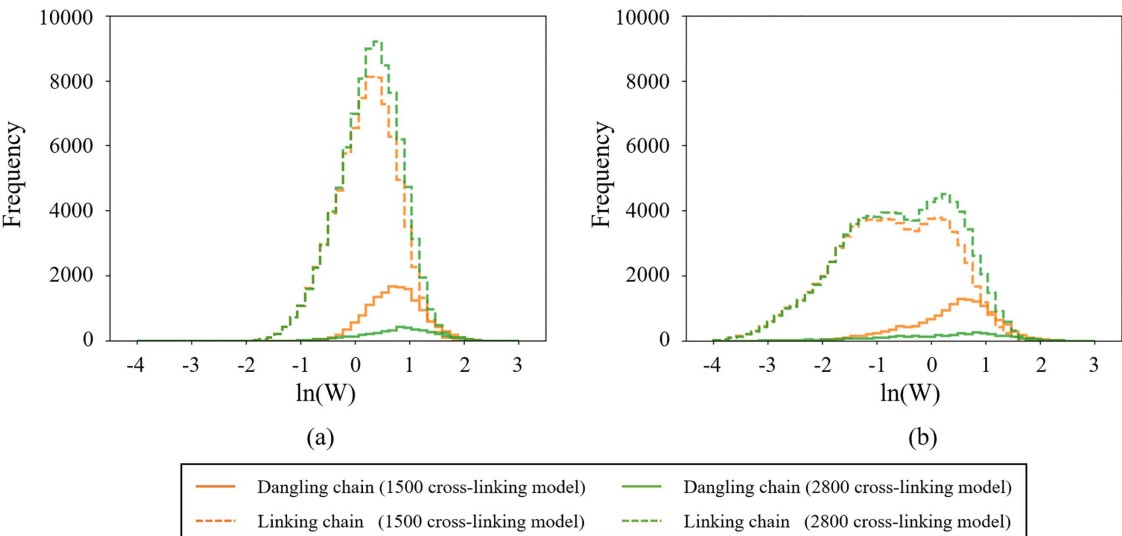

**Fig 6. The distributions of the NoS for the dangling chain and the network strand.** (a) The distributions at a stretch ratio of 1, (b) the distributions at a stretch ratio of 3.

leading to decrease in their NoS. Fig 7c shows the comparison with the distributions of the NoS of randomly extracted atoms from the network strand except for the preferentially extended chains. The NoS of the randomly sampled network strands remains almost unchanged during the deformation, though the NoS of the preferentially extended network strands decreases. Therefore, even in the entropic elasticity region, we found that the NoS of the specific bonds relating to the failure at a larger strain region decreases. It implies that weak points in the network structure, which lead to failure, could be detected even at small strain deformation within the entropic elasticity region.

## Conclusion

We developed an effective method capable of measuring macroscopic entropy from the atomic fluctuations reproduced by MD simulations. The NoS of the system related to macroscopic entropy was calculated by integrating the NoS of all atoms in the system, corresponding to the volume of the phase space. To validate the proposed method, the entropy-temperature relationships and the entropy-stretch ratio relationships calculated from the trajectories of the CGMD models were compared with those derived from the representative entropy simulation methods and thermodynamics, respectively. The results confirmed the efficiency of the proposed method, as the values were approximately the same. The changes in the internal structure of the cross-linked polymer network during deformation were investigated as a use case for the proposed method. We found that the change in the NoS of the network strands is greater than that of the dangling chains. Furthermore, we found that the NoS of the specific network strands leading to failure, which extend in the larger strain region, decreases even in a small strain region exhibiting entropic elasticity.

Although the proposed method was applied to CGMD simulations in this study, it is worthy to mention that this method is not limited to CGMD simulations. In analyzing a full-atomic MD simulations, our method can reveal the influence of molecular-scale structures, such as functional groups, terminal-modified polymers, polymer sequences, and geometrical isomers, on entropy. However, analysis of discontinuous structures, such as sequence heterogeneity in polymers, will require improvements to the proposed method for both CGMD and fill-atomic MD. For the homopolymer example presented in this study, the assumption of independence between the phase spaces of individual particles does not impose any limitations. Nevertheless, because neighboring molecular structure clearly affects particle mobility based on the mass

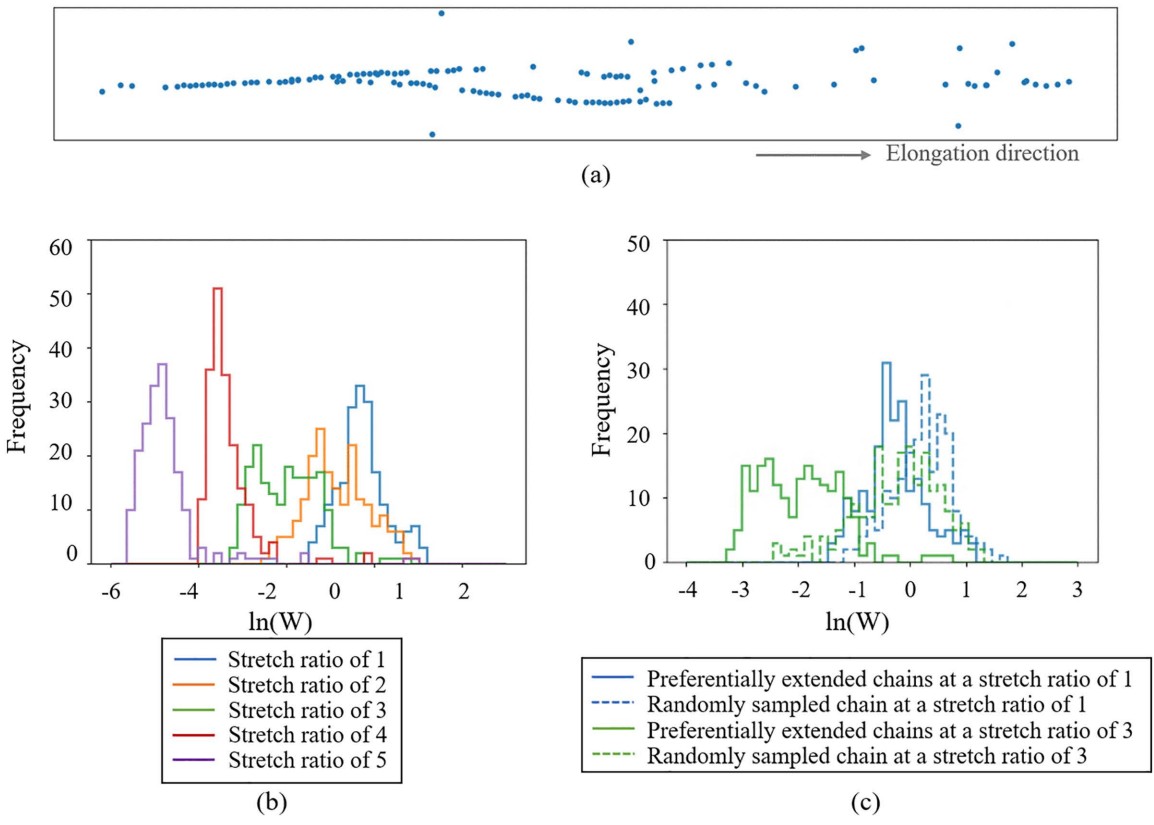

**Fig 7. History of the NoS for the extended chain at a stretch ratio of 5.** (a) Visualization of the extended chain at a stretch ratio of 5. The centers of each bond are visualized as circle markers. (b) The distributions of the NoS for the extracted atoms constituting the extended chains from a stretch ratio of 1 to a stretch ratio of 5. (c) Comparison of the NoS for the extracted atoms and the NoS for the randomly sampled atoms from the network strands at a stretch ratio of 1 and 3, which indicates entropic elasticity.

weight or stereo-structure, in regions where different structures are adjacent in copolymers, the proposed method can still be applied with either CGMD or full-atomic models to compute the NoS, but the local accuracy of the analysis may be reduced. Validation of this behavior and the development of mitigation strategies remain tasks for future work. Moreover, we aim to develop correction methods that consider higher-order correlations to improve computational accuracy. Additionally, the development of segmentation methods to reduce computational costs is also necessary. Furthermore, there is room for discussion regarding the investigation of the effects of the independence assumption. In terms of materials design, we aim to identify an ideal nano-meter scale structure to control macroscopic properties, such as toughness, elastic modulus, and viscoelasticity. Surprisingly, the proposed method is not limited to polymer materials; for instance, it can be employed in the design of lubricants dominated by the friction between metallic surfaces and base oils. Machine-learned interatomic potentials, which have recently attracted significant attention, can simulate chemical [64,65]. Thus, collaboration of the potentials with the proposed method can facilitate the analysis of changes in entropy and free energy during chemical reactions. We expect that our proposed method advances material science in various fields.

## Acknowledgments

We thank the researchers at ENEOS Materials Corporation for valuable discussions that contributed to the concept validation of this study.

## Author contributions

**Data curation:** Takafumi Ishii, Takashi Kojima.

**Supervision:** Yusuke Yasuda, Kazushi Fujimoto.

**Validation:** Takashi Kojima.

**Writing – original draft:** Takashi Kojima.

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
