## [Decision Letter · Decision Letter 0]

3 Mar 2026

Dear Dr. Kojima,

Thank you for submitting your manuscript to PLOS ONE. After careful consideration, we feel that it has merit but does not fully meet PLOS ONE’s publication criteria as it currently stands. Therefore, we invite you to submit a revised version of the manuscript that addresses the points raised during the review process.

We look forward to receiving your revised manuscript.

Kind regards,

Shaofeng Xu

Academic Editor

PLOS One

Journal Requirements:

https://journals.plos.org/plosone/s/file?id=wjVg/PLOSOne_formatting_sample_main_body.pdf and and and and https://journals.plos.org/plosone/s/file?id=ba62/PLOSOne_formatting_sample_title_authors_affiliations.pdf

[The authors have declared that no competing interests exist.].

We note that one or more of the authors are employed by a commercial company: ENEOS Holdings, Inc.

Reviewers' comments:

Reviewer's Responses to Questions

**Comments to the Author**

1. Is the manuscript technically sound, and do the data support the conclusions?

Reviewer #1: Yes

Reviewer #2: Yes

2. Has the statistical analysis been performed appropriately and rigorously?

Reviewer #1: Yes

Reviewer #2: Yes

3. Have the authors made all data underlying the findings in their manuscript fully available?

Reviewer #1: Yes

Reviewer #2: Yes

4. Is the manuscript presented in an intelligible fashion and written in standard English?

Reviewer #1: Yes

Reviewer #2: Yes

Reviewer #1: Review report of Manuscript PONE-D-25-68334

The authors have provided a method to calculate local entropy through the number of states for each atom within its phase space consisting of the position and momentum parameters in polymer networks. They have compared the entropy-temperature relationships and the entropy-stretch ratio relationships calculated from the trajectories of the coarse-grained MD models with theoretical predication.

In general, their results and the relative analysis seem reasonably reliable in this work and are of interest to experts in the research field of statistical physics. In my opinion, it surely merits publication in PLOS ONE. Pending a few comments that I hope could be helpful for the authors to improve their work.

1. First, the authors have made an assumption of independence between atomic phase spaces. The assumption that atomic phase-space volumes are independent is a significant simplification. The authors have not quantified its validity across different length scales or polymer architectures. At very least, the authors should provide a discussion about at what condition this assumption is fulfilled.

2. This method is relied on the statistical sampling of atomic fluctuations to compute phase-space volumes. However, the convergence of phase-space volumes with respect to the simulation length and ensemble size is not validated. In particular, the authors should give evidence for convergence at different temperature.

Furthermore, the authors have given results in the NVT ensemble. However, the entropy variation due to deformation is sensitive in the NVT ensemble. How the results would be in the NPT or NVE ensemble should be discussed.

3. The validation of this method in the coarse-grain model is acceptable. I am particularly concerned that whether this method could be applied in the All-atom models of other well-studied polymers (e.g., polyethylene, polystyrene). In particular, I am curious about how this model could estimate configuration-related entropy like the torsional entropy. The authors could add a discussion about this concern.

Reviewer #2: This paper proposes a method to estimate the entropy of local polymer-network structures by calculating the number of states (NoS) based on atomic fluctuations obtained from coarse-grained molecular dynamics simulations. The authors validate their method by comparing the results with thermodynamic principles and analyze the entropy changes of network strands and dangling chains during deformation. The study provides insights into the relationship between microscopic atomic fluctuations and macroscopic material properties.

The manuscript is generally well-written and presents an interesting approach to analyzing local entropy in polymer networks. The validation process and the application to network strands versus dangling chains are logically presented. I believe this work is suitable for publication after the authors address the following specific points to clarify the presentation of the data.

1. In Figure 6, the authors compare the distributions of the NoS for dangling chains and network strands at stretch ratios of 1 and 3 (Page 16). However, there is no mention or data presented for stretch ratios 2, 4, and 5 in the main text. It would be beneficial for the readers if the authors could add a brief explanation in the main text regarding why these specific stretch ratios were selected for comparison and why the others (particularly 2, 4, and 5) are not included in this figure.

2. Figure 7(a) visualizes the extended polymer chains at a stretch ratio of 5 (Page 17). However, the distributions of the NoS shown in Figure 7(b) only cover the range from stretch ratio 1 to 4 (Page 16, 17). The authors should address this discrepancy in the text. Specifically, please explain why the distribution at stretch ratio 5 is not included in Figure 7(b), despite it being the condition used to identify the extended chains in Figure 7(a).

I also noted a few minor errors and typos that should be corrected, as detailed below:

Page 14, Line 206: "In addition, In addition," - Please remove the duplicate phrase.

Page 16, Line 266: "in the elastic elasticity" - This phrasing seems incorrect. Did the authors mean "energetic elasticity" or simply "elastic region"?

Page 16, Line 267: "...likely to occur, extracted based on..." - It appears a verb is missing. It should likely read "...were extracted based on...".

Page 16, Line 268: "from at a stretch ratio of 1 to at a stretch ratio of 4" - The phrasing "from at... to at..." is awkward. Consider revising to "from a stretch ratio of 1 to 4".

Page 17, Line 284: "...where indicating entropic elasticity." - This should be corrected to "...which indicates entropic elasticity" or similar.

.

Reviewer #1: **Yes:** Jige ChenJige ChenJige ChenJige Chen

Reviewer #2: No

---

## [Author Response · Author response to Decision Letter 1]

11 Mar 2026

--------- Reviewer 1 ---------

1. Reviewer comment:

"First, the authors have made an assumption of independence between atomic phase spaces. ... At very least, the authors should provide a discussion about at what condition this assumption is fulfilled."

Response:

Thank you for this important point. For homopolymers with uniform structure, we consider that the proposed method has no inherent length-scale limitation (e.g., polymer chain length). Nevertheless, as discussed in the manuscript (Figure 5), the phase-space volume of each atom is affected by neighboring particles. When neighboring particles are of the same species, those influences are mutually similar; in that case the method may slightly overestimate some contributions, but we expect the quantitative results to remain reliable as discussed in Figure 5. In contrast, in heterogeneous systems such as copolymers, boundary particles that connect to a different species may have phase-space volumes that differ from particles in continuous same-species regions. Thus, applying the proposed method to strongly heterogeneous materials may require methodological improvements. Since this paper is intended as a first report and a proof-of-concept, we have designated such extensions as future work and added the discussion to the Conclusion section.

2. Reviewer comment:

"This method is relied on the statistical sampling of atomic fluctuations to compute phase-space volumes. However, the convergence ... is not validated. In particular, the authors should give evidence for convergence at different temperatures.”

Response:

Thank you for your comments. The sampling size and its temperature dependence are discussed in Figure 2. In order to make this clearer we modified the figure title.

2. “Furthermore, the authors have given results in the NVT ensemble. However, the entropy variation due to deformation is sensitive in the NVT ensemble. How the results would be in the NPT or NVE ensemble should be discussed."

Response:

As the reviewer pointed out, our deformation simulations were performed in the NVT ensemble. We confirmed that energy fluctuations during elongation process depend strongly on deformation speed and other simulation conditions. To obtain representative states for entropy evaluation we deform at a fixed rate, then fix the stretch ratio and relax the system until the total energy has converged; sampling is performed from that relaxed, energetically stable state (e.g. Figure 1). Because we sample from these energy-stable states, we expect equivalent results in the NVE ensemble. As an example, the energy history during sampling for the results shown in Figure 4(a). The energy is effectively stable during the sampling window. We have added this explanation and the example energy history to the manuscript.

3. Reviewer comment:

"The validation of this method in the coarse-grain model is acceptable. I am particularly concerned whether this method could be applied in the all-atom models of other well-studied polymers ... how this model could estimate configuration-related entropy like the torsional entropy."

Response:

Thank you for your comments. As indicated in equations (6) and (7) of the manuscript, by applying appropriate molecular-weight-based weighting the proposed approach can be extended to all-atom models. We have already applied related entropy measurements for polymers at inorganic-material interfaces (work in progress). However, presenting full all-atom results in this first paper would substantially increase the manuscript length and broaden the scope beyond the present proof-of-concept. Therefore, we confined this report to coarse-grained-model results and will present comprehensive all-atom applications, including discussion of torsional/configurational entropy contributions, in a subsequent paper.

--------- Reviewer 2 ---------

1. Reviewer comment:

"In Figure 6, the authors compare the distributions of the NoS for dangling chains and network strands at stretch ratios of 1 and 3 ... It would be beneficial ... if the authors could add a brief explanation ... why these specific stretch ratios were selected ... and why others (particularly 2, 4, and 5) are not included."

Response:

Thank you for your comments. As shown in Figure 4, entropic elasticity region is observed up to a stretch ratio of 3. To focus on changes of the NoS within the entropic elasticity region we compared results at stretch ratios 1 and 3. The result at stretch ratio 2 is essentially an interpolation between 1 and 3 and thus was omitted for clarity. We have added this explanatory sentence to the main text.

2. Reviewer comment:

"Figure 7(a) visualizes the extended polymer chains at a stretch ratio of 5 ... However, the distributions of the NoS shown in Figure 7(b) only cover the range from stretch ratio 1 to 4 ... Please explain why the distribution at stretch ratio 5 is not included in Figure 7(b)."

Response:

Thank you for pointing this out. We have now added the data at stretch ratio of 5 to Figure 7(b). The added result shows the expected further decrease of entropy at that deformation level.

3. Minor corrections (all fixed):

---

## [Decision Letter · Decision Letter 1]

24 Mar 2026

Dear Dr. Kojima,

Thank you for submitting your manuscript to PLOS ONE. After careful consideration, we feel that it has merit but does not fully meet PLOS ONE’s publication criteria as it currently stands. Therefore, we invite you to submit a revised version of the manuscript that addresses the points raised during the review process.

As the corresponding author, your ORCID iD is verified in the submission system and will appear in the published article. PLOS supports the use of ORCID, and we encourage all coauthors to register for an ORCID iD and use it as well. Please encourage your coauthors to verify their ORCID iD within the submission system before final acceptance, as unverified ORCID iDs will not appear in the published article. *Only* the individual author can complete the verification step; PLOS staff the individual author can complete the verification step; PLOS staff the individual author can complete the verification step; PLOS staff the individual author can complete the verification step; PLOS staff *cannot* verify ORCID iDs on behalf of authors.verify ORCID iDs on behalf of authors.verify ORCID iDs on behalf of authors.verify ORCID iDs on behalf of authors.

We look forward to receiving your revised manuscript.

Kind regards,

Shaofeng Xu

Academic Editor

PLOS One

Journal Requirements:

Reviewers' comments:

Reviewer's Responses to Questions

**Comments to the Author**

Reviewer #1: All comments have been addressed

Reviewer #2: All comments have been addressed

2. Is the manuscript technically sound, and do the data support the conclusions?

Reviewer #1: Yes

Reviewer #2: Yes

3. Has the statistical analysis been performed appropriately and rigorously?

Reviewer #1: Yes

Reviewer #2: Yes

4. Have the authors made all data underlying the findings in their manuscript fully available?

Reviewer #1: Yes

Reviewer #2: Yes

5. Is the manuscript presented in an intelligible fashion and written in standard English?

Reviewer #1: Yes

Reviewer #2: Yes

Reviewer #1: 2nd Review Report PONE-D-25-68334R1

The revised version of the manuscript is improved relative to the original version. I am satisfied with this revised version and their replies to my concerns. In particular, their simulation details and convergence validation are provided.

It is therefore my opinion that the manuscript is suitable for publication in its present form.

Reviewer #2: The authors have satisfactorily addressed most of my previous comments.

However, the figure legend for Figure 7 should be amended to align with the revised figure.

If these corrections are made, I recommend accepting the manuscript.

.

Reviewer #1: **Yes:** Jige ChenJige ChenJige ChenJige Chen

Reviewer #2: No

---

## [Author Response · Author response to Decision Letter 2]

25 Mar 2026

The figure legend for Figure7(b) was changed from “The distributions of the NoS for the extracted atoms constituting the extended chains from a stretch ratio of 1 to a stretch ratio of 4.” to ““The distributions of the NoS for the extracted atoms constituting the extended chains from a stretch ratio of 1 to a stretch ratio of 5.”

---

## [Editor Report · Decision Letter 2]

8 Apr 2026

An Analysis Method of Local Entropy Changes from Atomic Fluctuations.

PONE-D-25-68334R2

Dear Dr. Kojima,

We’re pleased to inform you that your manuscript has been judged scientifically suitable for publication and will be formally accepted for publication once it meets all outstanding technical requirements.

Kind regards,

Shaofeng Xu

Academic Editor

PLOS One
---

## [Editor Report · Acceptance letter]

PONE-D-25-68334R2

PLOS One

Dear Dr. Kojima,

I'm pleased to inform you that your manuscript has been deemed suitable for publication in PLOS One. Congratulations! Your manuscript is now being handed over to our production team.

Kind regards,

on behalf of

Dr. Shaofeng Xu

Academic Editor

PLOS One